# The Influence of Seed Characteristics on Seed Dispersal Early Stages by Tibetan Macaques

**DOI:** 10.3390/ani12111416

**Published:** 2022-05-31

**Authors:** Hanrui Qian, Wenbo Li, Jinhua Li

**Affiliations:** 1School of Resources and Environmental Engineering, Anhui University, Hefei 230601, China; x20201020@stu.ahu.edu.cn (H.Q.); lwb@ioz.ac.cn (W.L.); 2International Collaborative Research Center for Huangshan Biodiversity and Tibetan Macaque Behavioral Ecology, Anhui University, Hefei 230601, China; 3Institute of Zoology, Chinese Academy of Sciences, Beijing 100101, China; 4School of Life Sciences, Hefei Normal University, Hefei 230601, China

**Keywords:** seed dispersal, seed destruction, gut passage time, seed characteristics, Tibetan macaque (*Macaca thibetana*)

## Abstract

**Simple Summary:**

Seed dispersal by frugivores is critical to forest regeneration. However, the Tibetan macaque’s seed dispersal function and the effect of seed physical characteristics on seed dispersal effectiveness need to be confirmed. We not only demonstrated that Tibetan macaques could act as seed dispersers, but we also investigated the effect of seed physical characteristics on gut passage time and seed damage rates. Smaller seeds are less likely to be damaged and remain for longer periods of time, whereas larger and heavier seeds are defecated more quickly. In our experiments, the seed-to-shell investment rate had no effect on the damage rate. Understanding the role of seed dispersal in primates can aid in the understanding of seed dispersal symbiosis.

**Abstract:**

There are numerous ecological and evolutionary implications for the ability of frugivores to predate on fruits and consume or disperse their seeds. Tibetan macaques, which are considered important seed predators, typically feed on fruits or seeds. However, systematic research into whether they have a seed dispersal function is still lacking. Endozoochory allows seeds to disperse over greater distances by allowing them to remain in the animal’s digestive tract. Consumption of fruit may not imply effective seed dispersal, and the physical characteristics of seeds (e.g., size, weight, specific gravity, etc.) may influence the dispersal phase’s outcome. We conducted feeding experiments with three captive Tibetan macaques (*Macaca thibetana*) and nine plant seeds to determine the influence of seed characteristics on Tibetan macaques’ early stages of seed dispersal. The results revealed that the percentage of seed destruction (*PSD*) after ingestion was 81.45% (range: 15.67–100%), with the *PSD* varying between plant species. Among the three passage time parameters, the transit time (*TT*) (mean: 18.8 h and range: 4–24 h) and the time of seed last appearance (*TLA*) (mean: 100.4 h and range: 48–168 h) differed significantly between seed species, whereas the mean retention time (*MRT*) (mean: 47.0 h and range: 32–70.3 h) did not. In terms of model selection, *PSD* was influenced by seed size, weight, volume, and specific gravity; *TT* was influenced by seed-to-shell investment rate, weight, volume, and specific gravity; and *TLA* was influenced only by seed size. These findings imply that seeds with a smaller size, specific gravity, volume, and greater weight pass more easily through the monkeys’ digestive tracts. Particularly, seeds with a mean cubic diameter (*MCD*) of <3 mm had a higher rate of expulsion, larger volume, and weight seeds pass faster, while smaller remained longer. Tibetan macaques, as potential seed dispersers, require specific passage time and passage rates of small or medium-sized seeds. Larger and heavier seeds may be more reliant on endozoochory. Tibetan macaques have the ability to disperse seeds over long distances, allowing for gene flow within the plant community.

## 1. Introduction

Frugivores seed dispersal is the most extensive reciprocal mechanism of animal-plant interaction [1,2]. Frugivores disperse seeds primarily through endozoochory [3], which allows seeds to remain within the animal for a period of time before being transferred to the defecation site. Endozoochory allows seeds to travel greater distances [4]. Moreover, the scarification effect of the seed coat by the digestive tract [5] and nutrients provided by the organic matter in the feces [6] may promote seed germination and growth. This is an important way to promote forest regeneration [6,7,8].

According to the “Escape Hypothesis”, seeds that have escaped the vicinity of the parent tree are more likely to reproduce successfully than those that fall nearby [9,10,11]. This enhances the gene flow of plant species within the community. Thus, seed dispersal involves multiple levels of the organization, such as genes, populations, and communities, and has critical ecological and evolutionary implications for the ecosystems [1,12,13]. In some degraded Asian forests, primates are the most common consumers and dispersers of fruits that lack large frugivorous animals [14,15]. They play an important role in the dispersal of large fruits [16]. Cercopithecine monkeys are more efficient at using fragmented and degraded habitats than other primates [17,18], making them irreplaceable for forest regeneration and recruitment. Macaque (*Macaca* spp.) has an extensive geographic distribution within this order [19]. Researchers have investigated the involvement of many macaques as seed dispersers from diverse perspectives, including long-tailed macaque (*Macaca fascicularis*) [20], Japanese macaque (*M. fuscata*) [4,21,22,23], northern pig-tailed macaque (*M. leonina*) [24], and rhesus macaque (*M. mulatta*) [25], since the 1990s [26]. However, still, the understanding is not sufficient about the dispersal of the seeds after ingestion by the Tibetan macaque.

The contribution of primates to forest regeneration (Seed dispersal effectiveness: SDE) includes quantitative as well as qualitative aspects [27,28]. Here, the quantity denotes the total number of seeds that monkeys take and carry, depending on the feeding behavior [28]. In contrast, the quality is the probability of seed germination and establishment, which depends on seed destruction, gut passage time, dispersal distance, and germinability [17,28,29]. The gut passage times and movement patterns might affect seeds’ spatial distribution and germinability [30]. The physical characteristics of the seeds further affect their retention time in the digestive tract [30,31]. Some studies have shown that the effect of seed characteristics on passage time differs significantly among the primates. For instance, research on black howler monkeys (*Alouatta caraya*) and golden snub-nosed monkeys (*Rhinopithecus roxellana*) has shown that seed size is independent of the passage time [32,33]. However, smaller seeds have a longer passage time in woolly monkeys (*Lagothrix lagotricha*) [34]. The gut retention time of Japanese macaques (*M**. fuscata*) was positively correlated with seed weight and specific gravity [22]. Contrary to this, the gut retention time of two callitrichid species (*Saguinus mystax*
*and S. fuscicollis*) was negatively correlated with seed-specific gravity [35]. The above studies suggest that the relationship between seed characteristics and passage time is ambiguous and research is needed for better understanding and clarity. In addition, the body weight hypothesis suggests that gut passage time varies with the body weight [36], whereas daily traveled distance and home range of the mammalian species are known to increase with the body size [37]. The Tibetan macaque is the largest species in the genus of *Macaca* [38] and may have a greater positive impact on the seed dispersal effectiveness.

We used captive Tibetan macaque to explore the role that they play in seed dispersal. We determined whether the physical characteristics of the seeds, including size, volume, weight, specific gravity, and seed-to-shell investment rate, have an impact on their endozoochory. Smaller-sized seeds may be more advantageous in being ingested intact by the monkeys. The higher percentage of seeds invested in the shell may have a positive effect on resisting abrasion and erosion in the digestive tract of monkeys. We predicted that (1) seed damage rates varied by seed size; (2) the smaller the seed, the less likely it would be destroyed in the gut, while the higher the percentage of seeds invested in the shell, the less likely it would be harmed; (3) seeds have a specific retention period in the digestive tract, with smaller seeds remaining longer in the digestive tract and larger and heavier seeds being defecated more quickly.

## 2. Materials and Methods

### 2.1. Study Subjects

We conducted feeding experiments with three captive Tibetan macaques at the Hefei Wildlife Park (31°50′ N, 117°10′ E) in Anhui province, east China. All three individuals (age > 10 years; body weight: 16.3 ± 1.3 kg) were reared in the same cage (W 3.0 m × L 7 m × H 4 m). The cage consists of concrete walls and iron mesh, and only one door can be used to provide food for the monkeys. All three individuals were in similar physical condition and were able to move freely in a cage. The daily rations for all monkeys normally consisted of vegetables and fresh fruits (including tangerines, apples, sweet potatoes, bananas, and pumpkin chunks), peanuts, corn kernels, and water. During the trial periods, to avoid affecting the gut retention time, we entered the cages alone to minimize stress on the monkeys [22]. The monkeys were fed daily at 9:00. The fresh weight of the recipe food provided to each individual was controlled between 1000 and 1200 g. We collected enough mature seeds of nine species (*Syzygium buxifolium*, *Lindera glauca*, *Litsea pungens*, *Actinidia chinensis*, *Celastrus orbiculatus*, *Rosa laevigata*, *Akebia trifoliata*, *Kadsura longipedunculata*, *Viburnum dilatatum*) with varied dimensions, and all seeds were sourced from the field (Appendix A). These selected seed species are an important part of the diet of Tibetan macaques in the Tianhu Mountain Group of the Huangshan Mountains (Li et al. unpublished data).

### 2.2. Seed Characteristics

Before the experiment, we measured the size (length, width, and height) of 30 randomly selected seeds per species with a digital vernier caliper to the nearest 0.01 mm. Meanwhile, we also weighed 30 randomly selected dry seeds with an electric balance to the nearest 0.01 mg. They were then subsequently broken to measure the shell weight. The *Volume* of each seed was measured according to the formula proposed by Garber [35]:$$V=\pi R^{2}\left(L-\frac{2}{3}R\right),$$
where *R* = (seed width + height)/4, and *L* = seed length. The specific gravity (*SG*) is calculated by dividing the dry weight of the seed by its volume. The shell weight ratio (*SWR*) was obtained as (shell weight/dry seed weight). This variable represents the percentage of seed-to-shell investment. The mean cubic diameter (*MCD*) of seeds was calculated based on *MCD* = (*a*_1_ × *a*_2_ × *a*_3_)*^1/3^* [39]. Here *a*_1_, *a*_2_, *a*_3_ are replaced by the length, width, and height of seeds. All information about physical characteristics such as size, *Weight*, *Volume*, *SG*, *SWR*, *MCD* is displayed in Appendix A.

### 2.3. Feeding Experiments

We conducted five replicated experiments on nine seed species from early October to mid-December 2021. The cage conditions and daily rations of Tibetan macaques have not changed throughout the experiment. We fed a total of 10,130 seeds to the monkeys, which included 670 *Syzygium buxifolium*, 1040 *Lindera glauca*, 830 *Litsea pungens*, 1500 *Actinidia chinensis*, 1150 *Celastrus orbiculatus*, 1240 *Rosa laevigata*, 1250 *Akebia trifoliata*, 1200 *Kadsura longipedunculata*, 1250 *Viburnum dilatatum*, respectively (Appendix A). Monkeys were unable to ingest seeds directly. We embedded the seeds in blocks of bananas that are a fruit included in their daily rations and controlled the number of bananas provided to the monkeys to 3 or 4 per day, as redundant bananas may have some intestinal digestion-promoting effects. To avoid the impact of the size, shape, and other characteristics of the various types of seeds on the monkeys’ chewing and swallowing, we only gave the monkeys one type of seed every day. Before the same species were fed again, the remaining types of seeds were fed. Considering different types of seeds have distinct visual qualities, the above procedure has no effect on our ability to remove seeds from feces. The monkeys were provided with chunks of banana embedded with seeds before the typical daily rations resupply (at 9:00). The monkeys were able to consume the banana pieces in a short time (approximately 3 min), and no spitting out behavior was observed. Thus, we can consider that all the offered seeds were handled by the mouth and gut of the monkeys. According to our pre-experiments, the defecation of the three individuals was similar and infrequent. We also included the three individuals’ similar physical conditions as well as the need for the caging regulations that they could not be kept separately. As a result, we analyzed the feces using a combined analysis of the three individuals without considering individual differences. We collected feces every 4 h in the daytime. We did not collect the night feces because monkeys hardly ever defecate at night when they are resting (after 18:00). 

All feces were collected and filtered with water through a mesh sieve with three pore-size gradients (mesh size, 0.5 mm, 1.0 mm, and 2.0 mm) to extract any seeds that might be present. When we found the seeds in the feces, we recorded the fecal collection time, seed types and the number of seeds. We considered seeds with broken seed shells, those that had lost viability (softened seeds), or those not found in the feces as destructed. In a round of trials, we supposed that if a certain species of seed was not discovered in the feces more than 24 h, it was regarded entirely excluded. The next feeding round for a particular seed should take place after the previous round of that seed was considered to be entirely excluded. Based on the data we obtained, we calculated the percentage of seed destruction (*PSD*). We replace the passage time of the seed with three parameters: time of the first appearance of a seed, which is the transit time (*TT*), time of the last appearance of a seed (*TLA*), and mean retention time (*MRT*). The *MRT* was calculated according to the following formula used by Lambert et al. [40]:$$MRT=\overset{n}{\underset{i=1}{\sum }}m_{i}t_{i}/\overset{n}{\underset{i=1}{\sum }}m_{i}$$
where $m_{i}$ is the number of seeds defecated at time $t_{i}$ after ingestion by the Tibetan macaque.

### 2.4. Statistical Analyses

We tested the effect of seed species on the percentage of seed destruction (*PSD*) and on the three variables linked to passage times (*TT*, *MRT*, *TLA*). To explore the differences in passage times between seed species, we removed three seed species (*Syzygium buxifolium*, *Lindera glauca*, and *Litsea pungens*) because they were barely collected intact (0.00% ± 0.00%, 0.17% ± 0.33%, and 0.21% ± 0.26%) during the five rounds of the trial. In our data, *TLA* satisfied the conditions of one-way ANOVA. While *TT* and *PSD* failed to satisfy normality (tested by Shapiro–Wilk normality test, *p* < 0.05), *MRT* test result for equal variances were less than 0.05 (*sig* = 0.046). Therefore, to reach our analysis, the non-parametric Kruskal–Wallis analyses of variance (ANOVAs) were applied to the last three variables. We analyzed the correlation between the three passage time variables using Spearman’s correlation analyses. We then depicted the distribution of gut passage times for different seed species ingested by captive Tibetan macaques. We chose an information-theoretic approach that provided methods for model selection and (multi) model inference [41] to assess seed characteristics (*Volume*, *MCD*, *Weight*, *SG*, and *SWR*, Table 1) that were most important for the *PSD* and the passage times of ingested seeds in caged Tibetan macaque. We created a full model that contains five characteristic variables that could potentially influence the results. This full model contained linear models built with all possible subsets of these five variables. We used the Akaike Information Criterion corrected for small sample sizes (AIC_C_) to select the “best” model. Models that were within two units of the smallest AIC_C_ value were considered to be credible (∆AIC_C_ < 2). Ultimately, we used the AIC_C_ values and Akaike weights that are viewed as the probability that the model is the best model among all considered models to identify the best model. Simultaneously, to make inferences about various predictors among all possible models with relative weights considered, we used multi-model inference to assess the relative influence of the variables. Only one of the models we used to explain the effect of physical characteristics of seeds on *PSD* had an ∆AIC_C_ < 2. As a result, we considered the top-level model to be the best model (*Volume + MCD + Weight + SG*, *w_i_* = 0.78), with variables that had a potential impact on *PSD* included in it, and there were three and five possible models selected by ∆AIC_C_ < 2 that have potential effects on the physical characteristics of seeds on *TT* and *TLA*, respectively. The model weights were selected as the best models for the respective top-level models in *TT* (*Volume + Weight + SG + SWR, w_i_* = 0.35) and *TLA* (*MCD*, *w_i_* = 0.21) (Table 1). We combined the “*glmulti*” and “*MuMIn*” packages that provide the necessary functionality for model selection and multi-model inference to complete the above analysis. All analysis was used by R language version 4.0.3 (R Core Team, 2020). Significance levels were set at *p* = 0.05 for these analyses.

### 2.5. Ethical Note

The experimental methods used in this work were compliant with China’s legal requirements for animal welfare and protection.

## 3. Results

### 3.1. Damage to Seed

Among the 10,130 seeds we fed in the five replicated experiments, the total number of defecated seeds ranged from 0–1036 (*Litsea pungens*: 2, *Syzygium buxifolium*: 0, *Lindera glauca*: 2, *Akebia trifoliata*: 96, *Kadsura longipedunculata*: 82, *Rosa laevigata*: 154, *Viburnum dilatatum*: 452, *Celastrus orbiculatus*: 82, *Actinidia chinensis*: 1036) (Appendix A). In general, seeds through the gut had a high percentage of damage (n = 45, mean ± SD = 81.45 ± 22.48%; range = 15.67–100%). The species having the lowest damaged seeds was *Actinidia chinensis* (n = 5, mean ± SD = 30.93 ± 7.73%), and the highest damage rate was obtained by *Syzygium buxifolium* (n = 5, mean ± SD = 100.00 ± 0.00%) (Figure 1). Seeds with the *MCD* of less than 3 mm were also able to pass through more easily (*PSD* < 70%) (Appendix A). The *PSD* differed significantly between nine seed species (Kruskal–Wallis test: χ^2^ = 41.85, *df* = 8, *p* < 0.001; Figure 1).

### 3.2. Gut Passage Time

The mean values of the passage times (*TT*, *TLA*, and *MRT*) of the six seed species in the gut were 18.8 h (range: 4–24 h), 100.4 h (range: 48–168 h), 47.0 h (range: 32.0–70.3 h), respectively (Figure 2). The results showed a significant correlation between *MRT* and *TLA* (Spearman’s correlation analyses, *r_s_* = 0.763, *p* < 0.001), the rest were not (Spearman’s correlation analyses, *MRT* and *TT*: *r_s_* = 0.113, *p* = 0.553; *TLA* and *TT*: *r_s_* = 0.278, *p* = 0.137). The results of testing the differences in the passage time variables on seed species showed that *TT* (Kruskal–Wallis test: χ^2^ = 23.54, *df* = 5, *p* < 0.001) and *TLA* (one-way ANOVA: *F*_5,24_ = 2.789, *p* = 0.040) differed significantly among the seed species, but not *MRT* (Kruskal–Wallis test: χ^2^ = 10.810, *df* = 5, *p* = 0.055) on seed species (Figure 2). The temporal distribution of the passage of different seed species shows that 58–81% of the seeds were passed within 48 h, while up to 72 h, 84–98% of the seeds were passed (Figure 3). Seeds ingested by Tibetan macaques were defecated between 100 and 168 h after feeding (Figure 3).

### 3.3. Effects of Seed Characteristics on PSD and Passage Times

*MCD* (*w* = 1.000, *β* ± SE = 0.294 ± 0.021, *p* < 0.001), *Weight* (*w* = 1.000, *β* ± SE = −0.011 ± 0.002, *p* < 0.001), *SG* (*w* = 0.985, *β* ± SE = 0.728 ± 0.193, *p* < 0.001), and *Volume* (*w* = 0.984, *β* ± SE = 0.004 ± 0.001, *p* < 0.001) were the variables that affected *PSD*, *SWR* (*w* = 1.000, *β* ± SE = 55.570 ± 7.077, *p* < 0.001), *SG* (*w* = 0.997, *β* ± SE = −60.127 ± 18.295, *p* = 0.001), *Volume* (*w* = 0.844, *β* ± SE = −0.226 ± 0.127, *p* = 0.076), and *Weight* (*w* = 0.567, *β* ± SE = −0.220 ± 0.247, *p* = 0.372) were the variables that affected *TT*, and *MCD* (*w* = 0.827, *β* ± SE = −17.691 ± 12.734, *p* = 0.165) was the only variable that affected *TLA*, according to the AIC_C_ models and relative influence of the variables (Table 2).

## 4. Discussion

The analyses confirmed some of our predictions that seed damage rates differ significantly among the species. We noticed that smaller-sized seeds were less likely to become damaged and remain in the gut for a longer period, whereas larger, heavier seeds tended to be defecated more quickly. Unexpectedly, we observed in our experiments that the damage rate was not affected by the seed-to-shell investment rate. Our results indicate that more than half of the seeds were defecated just 24 h after the ingestion, while the longest passage time was greater than 168 h.

The quality of seed dispersal is influenced by several factors: the level of seed damage, the length of time that seeds remain in the gut, the spatial extent of their movements, and the ability of seeds to germinate after being defecated [27,28]. Our results specify that the Tibetan macaques are important seed predators that provide potential value for seed dispersal. The recovery rate of intact seed ingested by captive Tibetan macaque was 18.55% (range: 0.00–69.07%). Moreover, we noted that damage rates differ significantly among seed species. Bai et al. reported similar results (29.9%; range: 0.0–65.3%) with 12 different fruits fed to caged golden snub-nosed monkeys [32]. Similarly, another study indicated a 19.7% (range: 1.5–45.8%) seed recovery rate for the captive Southern muriquis (*Brachyteles arachnoides*) [42]. Therefore, the seeds showed similar intact rates after passing through the digestive tract of primates of three diverse families having a difference in the digestive system. This observation indicates that differences in gut or digestion may not be the main factor affecting seed damage. Some primates may have the tendency to drop out the seeds during fruit consumption or to spat out the seeds; however, it was not supported in our observation during the experiments. This could probably be due to the selected size range (1.4–6.2 mm) of the seeds. For instance, two primate species (*Cercopithecus mitis doggetti and C*. *l’hoesti*) tended to drop seeds that were larger than 10 mm [43]. In our results, evidence of seed predation could be obtained for species (*PSD*, range: 99.5–100.0%) having *MCD* > 4.5 mm. In contrast, seeds with *MCD* < 3 mm had a better passage rate (*PSD* < 70%). Our results are in agreement with the finding that seeds with *MCD* > 3 mm were predated [44]. Although large seeds were damaged at a high rate in our study, the encrustation of berries from genuine fruits may have a positive influence on intact seed ingestion by monkeys, and previous research has identified the genus *Macaca* as a reliable seed disperser [45]. Our findings could point to the possibility of Tibetan macaques dispersing seeds with an *MCD* of less than 3 mm. Our model results on seed physical characteristics also confirm that seeds having a smaller size, heavier weight, lighter specific gravity, and smaller volume pass through the gut more smoothly. Toward this approach, we introduced shell to weight ratio (*SWR*), which represents the percentage of seed-to-shell investment. Interestingly, we did not find it to have a significant impact on the *PSD*. This is because a large part of the damage to seeds comes from the monkeys’ mouths, such as chewing and abrasion. Further, the shells of the seeds that we selected did not reach a threshold that would make it difficult for the monkeys to chew. Similar evidence has been found in the studies that dental adaptation defines primate predation on the seeds more directly than the gut modification [31].

Seed passage time in the gut is a proxy for seed dispersal distance, while the distance is influenced by the interaction of passage time and disperser movement patterns [21]. In the current study, we tested *TT*, *MRT*, and *TLA* at approximately 18.8 h, 47.0 h, and 100.4 h, respectively. There are some similarities as well as variations between this study and other reports on the primates. *Hylobates lar*, *Trachypithecus auratus*, *Alouatta caraya*, and other macaques such as *M*. *cyclopis*, *M*. *fascicularis*, and *M*. *fuscata* have a shorter passage time than *M*. *thibetana* [22,33,46,47]. Orangutans (*Pongo pygmaeus*), having the largest body weight, have the longest passage time than other species. The next longest passage time is noted for the Tibetan macaque, the largest species in the genus *Macaca*. This implies that seeds may be carried farther away from the parent tree by Tibetan macaques. This result also confirms the previous finding that passage time increases with higher body size in the same group of mammals [36]. However, the influence of digestive physiological factors and seed traits also seems to be important. The longer passage time is compatible with a high-fiber diet [48].

In this study, *TT* (*p* < 0.001) and *TLA* (*p* = 0.040) exhibited substantial differences in selected seed species, however, *MRT* (*p* = 0.055) showed no significant changes. Furthermore, our findings show that *MCD* is the sole variable that has a negative impact on *TLA*. This implies that seeds with lower *MCD* can stay longer in the digestive tract, whereas seeds with higher volume and heavier weight can rapidly move through the digestive tract (*Volume*, *Weight,* and *SG* have a negative effect on *TT*) (see Table 2). Previous research found similar results. Smaller seeds were able to stay longer in the digestive tract of animals [34]. A negative relationship was shown between seed-specific gravity and passage time in the two species of callitrichid primates (*Saguinus mystax* and *Saguinus fuscicollis*) [35]. As smaller-sized seeds are more likely to be entrapped in the folds specialized digestive system, thereby prolonging the *TLA* of those seeds. However, Tsuji’s study on caged Japanese monkeys found an inverse link between specific gravity, weight, and passage time [22]. This could be related to the varied ranges of the variables we looked at: seed *Weight* ranged from 1.6–59.9 mg, and *SG* was 0.39–1.06 mg·mm^−3^ in our investigation (Appendix A), whereas *Weight* was 0.19–25.7 mg and *SG* was 0.44–3.33 mg·mm^−3^ in Tsuji’s experiment [22]. Moreover, it is to be noted that seeds with higher (>1.0) specific gravity (2.95, 3.33) were brought from two different sizes of plastic seeds of their choice. Furthermore, the shape and hardness of the seeds may also be the reason for the difference in the results. Our findings conform to previous research that seed size influences the passage time in the digestive tract of vertebrates (including primates) and that larger and heavier seeds have shorter passage times, resulting in shorter dispersal distances [30].

The home range of Tibetan macaques in the Tianhu Mountain Group of the Huangshan Mountains, China, calculated by the convex polygon method, is approximately 7.7 km^2^. During the peak fruiting periods (summer and autumn), their daily travel distance varies between 1.5 and 3.0 km (Li et al. unpublished data). This has an important implication for our understanding of the seed dispersal in the wild Tibetan macaque. The larger seeds, in most circumstances in a plant community, could only be dispersed to a few meters [49], resulting in the occurrence of genetic ‘neighbourhoods’ in closely related individuals of many plant species [37]. According to the results of the passing time distribution in caged Tibetan macaques (Figure 3), more than half of the seeds were defecated after 24 h. Seeds with longer passage time, such as *Actinidia chinensis* (168 h), may enhance the diversity of the locations where seeds were defecated and hence the opportunity for gene flow across the plant community. 

Our experiments on captive Tibetan macaques allow us to control conditions that are difficult to regulate in the field. At the same time, this is also a limitation because wild monkeys may influence gut passage times due to the changes in the metabolic rates caused by long daily movements. There are other factors, such as the potential that the diet of free-living monkeys may contain more fiber, which will influence gut passage time. Individual differences cannot be ignored, and individual health conditions may also have an impact on digestion and thus on the gut retention time. In this study, we considered male Tibetan macaques, while the females were not considered. Some studies have pointed out the difference between males and females [33], owing to metabolic changes caused by the female’s cycle of ovulation or lactation [50]. In our future work, we could explore the germinability of the seeds that pass through the digestive tract of the Tibetan macaque, which is a necessary process to understand the effectiveness of seed dispersal. Future research should include the fieldwork and not only the role of primate species but also the synergistic role of primates with other species in the seed dispersal network. For example, in forest ecosystems, the redundancy and complementarity between Tibetan macaques and other species in the seed dispersal networks is important [18], as well as the second dispersal by the dung beetles [51].

## 5. Conclusions

Our results specify that the Tibetan macaques are important seed predators that provide potential value for seed dispersal. Particularly, they may have a significant dispersion value for seeds that are smaller than 3 mm in diameter. To fill some of the gaps in primate seed dispersal studies, we tested *PSD* and seed passage times of captive Tibetan macaques after ingesting seeds, as well as the effect of physical characteristics of the seeds in their early stages of seed dispersal. Specific *PSD* and passage times suggest that Tibetan macaques could act as seed dispersers and that ingested seeds are potentially dispersed over long distances. Some physical characteristics of the seeds are also important factors affecting *PSD* and passage times. Further research consideration should be provided to the viability of seeds that have been dispersed by the primates and their role in the seed dispersal network. This holistic approach will determine the role of primates in seed dispersal symbiosis.

## Figures and Tables

**Figure 1 animals-12-01416-f001:**
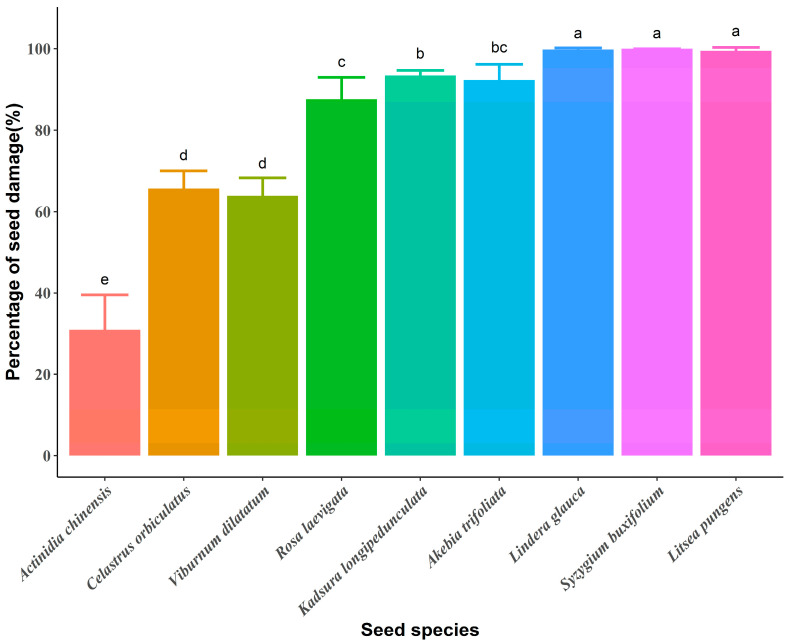
Percentage (mean % ± SD) of the destruction of nine seed species ingested through captive Tibetan macaques (*Macaca thibetana*) and the differences in the percentage of seed destruction (*PSD*) between seed species. The seeds are arranged according to *MCD* from smallest to largest. Different letters indicate significant differences (*p* < 0.05) among groups.

**Figure 2 animals-12-01416-f002:**
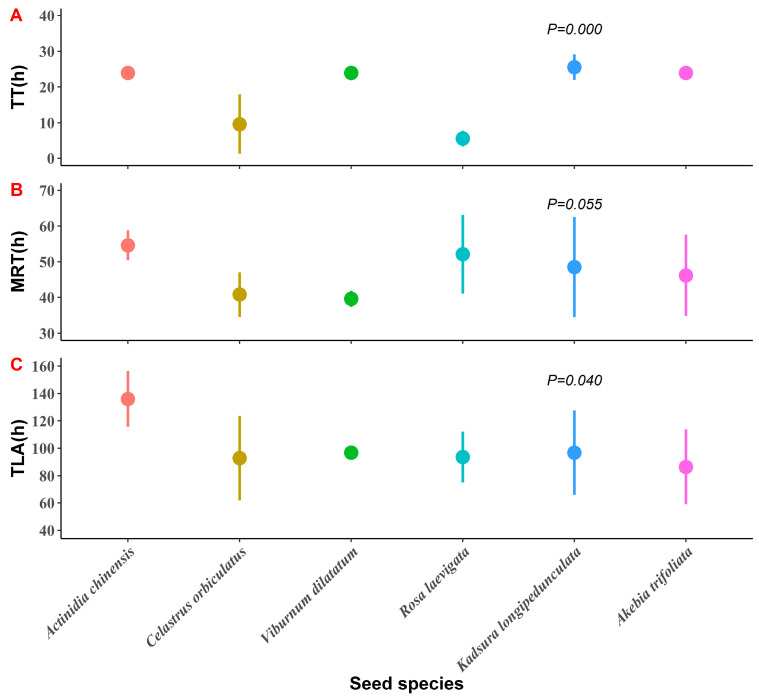
Relationships between seed species and three passage time parameters: (**A**): time of the first appearance of a seed (TT); (**B**): mean retention time (MRT); (**C**): time of the last appearance of a seed (TLA). The dots represent the mean and the bars show the standard deviation (SD).

**Figure 3 animals-12-01416-f003:**
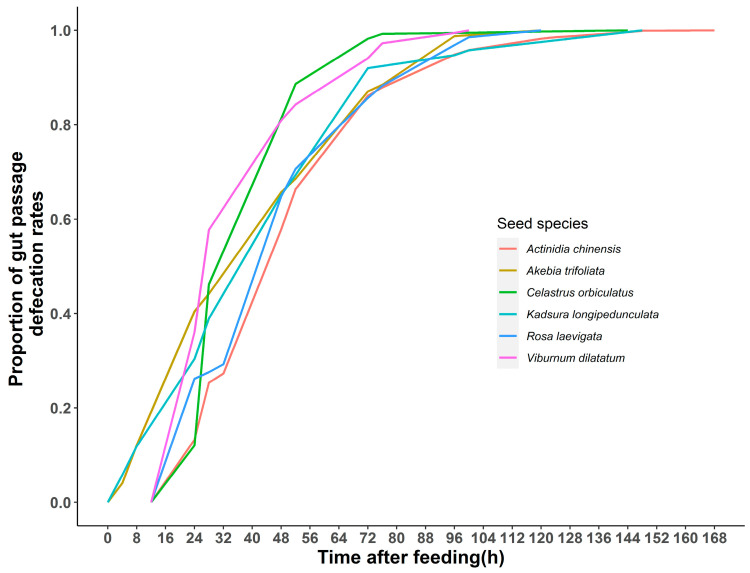
Cumulative distribution of gut passage time through the digestive tract of captive Tibetan macaques of six seed species. Gut passage rates are averaged across five rounds of experiments.

**Table 1 animals-12-01416-t001:** Model selection and performance measures for models used to explain the effect of different physical characteristics of seeds ingested by Tibetan macaques on *PSD* and passage time.

Model	K	AICC	∆AIC_C_	*wi*
*PSD*				
Volume + MCD + Weight + SG	6	−103.26	0.00	0.78
Volume + MCD + Weight + SG + SWR	7	−100.59	2.68	0.20
*TT*				
Volume + Weight + SG + SWR	6	174.14	0.00	0.35
Volume + MCD + SG + SWR	6	174.61	0.48	0.27
*TLA*				
MCD	3	281.00	0.00	0.21

The model was ranked by ∆AIC_C_ values from smallest to largest. K = number of parameters, AICC = Akaike’s information criterion values, ∆AIC_C_ = difference between the AICC value of the specified model and the optimal model, *wi* = model weight, *PSD* = percentage of seed damage, *TT* = time of first appearance of a seed, *TLA* = time of last appearance of a seed, *Volume* = volume of seeds, *MCD* = mean cubic diameter of seeds, *Weight* = dry weight of seeds, *SG* = specific gravity of seeds, *SWR* = shell weight ratio of seeds.

**Table 2 animals-12-01416-t002:** The inference and importance (*w*) of various predictors considered under all possible models were used to support the best model to account for the effect of different physical characteristics of seeds ingested by Tibetan macaques on PSD and passage time.

Variable	*w*	Estimate	SE	*z*	*p*
*PSD*					
Intercept	1.000	−0.715	0.203	−3.532	0.000 ***
MCD	1.000	0.294	0.021	13.740	0.000 ***
Weight	1.000	−0.011	0.002	−6.105	0.000 ***
SG	0.985	0.728	0.193	3.768	0.000 ***
Volume	0.984	0.004	0.001	3.619	0.000 ***
SWR	0.220	0.008	0.026	0.322	0.748
*TT*					
Intercept	1.000	53.213	14.774	3.602	0.000 ***
SWR	1.000	55.570	7.077	7.852	0.000 ***
SG	0.997	−60.127	18.295	−3.287	0.001 **
Volume	0.844	−0.226	0.127	−1.777	0.076
Weight	0.567	−0.220	0.247	−0.893	0.372
MCD	0.389	−0.584	1.492	−0.391	0.697
*TLA*					
Intercept	1.000	160.962	37.432	4.300	0.000 ***
MCD	0.827	−17.691	12.734	−1.389	0.165
Weight	0.396	0.326	0.853	0.382	0.702
Volume	0.296	−0.055	0.181	−0.301	0.763
SG	0.284	−9.809	22.951	−0.427	0.669
SWR	0.276	−7.401	15.593	−0.475	0.635

*w* = The importance value for a particular predictor is equal to the sum of the weights/probabilities for the models in which the variable appears. Usually importance value higher than 0.8 was considered an important variable. Significant differences: ***: *p <* 0.001, **: *p <* 0.01.

## Data Availability

The data presented in this study are available in the manuscript.

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
