# Peer review of "The Influence of Seed Characteristics on Seed Dispersal Early Stages by Tibetan Macaques"

_animals, 2022, doi:10.3390/ani12111416_

Round 1
Reviewer 1 Report
[General comments]
The present MS has following two big problems. I therefore judged this MS to be rejected.
1.The authors’ experimental design is not appropriate. Since their subject animals are housed together, the authors cannot obtain independent data. If they want to address effect of seed characteristics on the gastrointestinal passage time, they need to conduct corresponding experiments for individually caged animal. If the authors would like to conclude something with present results, they only can show describe the averaged value for each species.
- Description of this MS is very similar to that in previous paper (part of method section was almost full copy). The text should be revised so that it is not considered as plagiarism. Then, please cite previous paper(s) from which you designed the experiment.
[Specific comments]
Introduction
L89-96: The authors need to show the reason these prediction were raised.
L94: "suitable" Not "specific"?
Materials and Methods
I found that several sentence of this section is all-copy of previous paper (Tsuji et al. 2010 J. Zool. 280:171-176). The text should be revised so that it is not considered as plagiarism. Then, please cite previous paper(s) from which you designed the experiment.
L99-114: In this section, authors need to mention whether the conditions comply the animal welfare guidelines.
102: Did the three subject animals housed together? Could the authors discriminate who ate the banana chunks? And could they collect feces of target individual correctly? I understand that they just collected all feces on the ground together, which weaken the accuracy of the estimated values. Furthermore, were not there competition over food resources among them, and inter-individual difference in amount of feeding? You need to mention these point.
Then, in discussion section, you need to state what need to be improved.
L110-112: Do the macaques really feed on fruits of these plant species? You need to show the reference paper(s), with value of feeding percentages.
L124: ”shell weight ratio” In introduction, you need to show the background of the reason you included this variable in your analyses.
L130: "five replicated" No. In this study, no replication was performed, because three animals are housed together and you just feed them repeatedly. See comment below.
L157: You need to show definition of "seed destruction".
L166-208: Since three animals are housed together, your experimental design is the pseudo-replication (each experiments was not independent). If you want to address effects of seed characteristics on the seed destruction and gastro-intestinal passage time, you needed to conduct the feeding experiments for individually housed animals. Unfortunately, you cannot discuss this point in this paper, and just describe the averaged values (n = 1 for each species). Only seed size - passage time Relationship can be performed.
Results
As I mentioned above, your experimental design is wrong, and analyses for effect of seed characteristics on passage times need to be reconstructed. Thus, I do not comment.
Discussion
L275-282: In this part, you need to compare your results with those obtained from other macaques. Following paper might be useful.
Tsuji et al. (2011) Mamm. Biol. 76:525–533
Sengupta et al. (2014) Am. J. Primatol. 76: 1175-1184.
L316-337: See my comments above. You cannot discuss it, due to your wrong experimental design.
Reviewer 2 Report
Overall, this manuscript has average merit and is a very preliminary assessment of the role Tibetan Macaques may play in seed dispersal. The study size, and the difference in diet and activity, makes comparing wild and captive macaques in this context borderline. The authors do acknowledge this fact in the discussion, which is appropriate. There are moderate language modifications needed in both the abstract and the introduction:
Line 29 has a typo; predation is in the incorrect tense.
Line 60/61 does not seem complete.
Line 81 also has a typo and perhaps missing words
Reviewer 3 Report
The study investigates the possible role that Tibetan macaques may play as seed dispersal agents by investigating the degree of damage done to seeds of a range of species during gut passage following ingestion. The study itself is sound and the methodology is appropriate. There are some methodological points which require clarification however. The authors are advised to have the document proof read by a first language English speaker prior to submission because this might clarify some of the methodological issues for the reader.
The only issue which stands out is the conclusion drawn by the authors that their study provides evidence that Tibetan macaques act as agents of seed dispersal. As outlined in the feedback, the data presented support the notion that this species may be an effective seed dispersal agent, given that seeds are able to pass through the guts of individuals, but the authors cannot reasonable state that the evidence shows that the species is an agent of seed dispersal. The data are not appropriate to draw this conclusion.
With all this said, I recommend that the study be accepted for publication following minor revision.
